# Breast Cancer Risk Assessment Tools for Stratifying Women into Risk Groups: A Systematic Review

**DOI:** 10.3390/cancers15041124

**Published:** 2023-02-09

**Authors:** Louiza S. Velentzis, Victoria Freeman, Denise Campbell, Suzanne Hughes, Qingwei Luo, Julia Steinberg, Sam Egger, G. Bruce Mann, Carolyn Nickson

**Affiliations:** 1The Daffodil Centre, The University of Sydney, A Joint Venture with Cancer Council NSW, Sydney, NSW 2011, Australia; 2Melbourne School of Population and Global Health, University of Melbourne, Carlton, VA 3010, Australia; 3Centre for Outcomes Research and Effectiveness, Research Department of Clinical, Educational & Health Psychology, University College London, London WC1E 7HB, UK; 4Breast Service, The Royal Women’s and Royal Melbourne Hospital, Parkville, VIC 3010, Australia; 5Department of Surgery, University of Melbourne, Parkville, VIC 3010, Australia

**Keywords:** risk prediction models, breast cancer screening, risk assessment, risk-based screening

## Abstract

**Simple Summary:**

Early detection of breast cancer in asymptomatic women through screening is an important strategy in reducing the burden of breast cancer. In current organized breast screening programs, age is the predominant risk factor. Breast cancer risk assessment tools are numerical models that can combine information on various risk factors to estimate the risk of being diagnosed with breast cancer within a certain time period. These tools could be used to offer risk-based screening. This systematic review assessed, using a variety of methods, how accurately breast cancer risk assessment tools can group women eligible for screening within a population, into risk groups, so that each group could potentially be offered a screening protocol with more benefits and less harms compared to current age-based screening.

**Abstract:**

Background: The benefits and harms of breast screening may be better balanced through a risk-stratified approach. We conducted a systematic review assessing the accuracy of questionnaire-based risk assessment tools for this purpose. Methods: Population: asymptomatic women aged ≥40 years; Intervention: questionnaire-based risk assessment tool (incorporating breast density and polygenic risk where available); Comparison: different tool applied to the same population; Primary outcome: breast cancer incidence; Scope: external validation studies identified from databases including Medline and Embase (period 1 January 2008–20 July 2021). We assessed calibration (goodness-of-fit) between expected and observed cancers and compared observed cancer rates by risk group. Risk of bias was assessed with PROBAST. Results: Of 5124 records, 13 were included examining 11 tools across 15 cohorts. The Gail tool was most represented (*n* = 11), followed by Tyrer-Cuzick (*n* = 5), BRCAPRO and iCARE-Lit (*n* = 3). No tool was consistently well-calibrated across multiple studies and breast density or polygenic risk scores did not improve calibration. Most tools identified a risk group with higher rates of observed cancers, but few tools identified lower-risk groups across different settings. All tools demonstrated a high risk of bias. Conclusion: Some risk tools can identify groups of women at higher or lower breast cancer risk, but this is highly dependent on the setting and population.

## 1. Introduction

Early detection of breast cancer in asymptomatic women through screening is an important strategy in reducing the burden of breast cancer. Mammographic screening programs have decreased mortality for screened women and reduced the intensity of breast cancer treatment and associated sequelae [1,2,3,4]. Nevertheless, breast screening also confers potential harms such as overdiagnosis leading to the treatment of tumours that would not have progressed to symptomatic disease within a person’s lifetime, and false positive screening tests, associated with adverse psychological effects and possible reduced screening reattendance [5]. Current organised breast screening programs are directed to specific age groups, so that age is the predominant risk factor [3,6,7,8]. However, there are numerous other risk factors for breast cancer. More personalised, risk-based approaches to screening are expected to improve the balance of benefits and harms for identified risk groups [9,10]. This would require a rigorous and reliable method to routinely assess breast cancer risk in screening populations.

Breast cancer risk assessment tools (also known as risk prediction models) use numerical models to combine information on various risk factors (or risk predictors) to estimate the risk or probability of being diagnosed with breast cancer within a certain time period (e.g., 5 or 10 years) or from the time of assessment to older age [11]. These tools have evolved over time. Where earlier risk assessment tools considered information on reproductive factors (e.g., age at menarche/menopause, age at first live birth), family history, and breast biopsies, later tools incorporated additional lifestyle information (e.g., menopausal hormone therapy, alcohol consumption, smoking), anthropometric data (weight, height), ethnicity or/and mammographic density and various more recent tools incorporate genetic information in the form of polygenic risk scores (PRS) from analysing single-nucleotide polymorphisms associated with inherited variance in breast cancer risk [12,13]. Highly penetrant (“pathogenic”) variants in BRCA1/2 or other key genes are also included in some tools, e.g., Tyrer-Cuzick and BODICEA.

While many of these tools have been developed for individual clinical applications or management of higher-risk population groups, such risk assessment tools could potentially be used to stratify screening populations into population-level risk groups, with each group offered a screening protocol to optimise the benefits and minimise the harms of screening [14]. In line with increasing interest in personalised medicine and risk-based screening over the last decade [15,16] there has been a growth in publications concerning breast cancer risk assessment tool development, validation and evaluation. The wealth of tools now available are not widely utilised for the general population mainly due to insufficient validation, lack of available resources for capturing complete risk factor information from screening participants and the need to agree on, and resource, tailored screening protocols for specific risk groups [17,18]. 

A critical step making the most of available tools is understanding which tools can accurately achieve population-level risk stratification, including the extent to which their accuracy can be generalised to different populations and health settings. Case-control studies frequently report improvements in the discrimination of new or revised risk assessment tools [12,19,20,21]; however, risk assessment tools can only be adequately assessed for the purpose of population-level implementation when they are externally validated on populations different to the study groups on which they were developed. 

This systematic review aims to characterise studies which compare breast cancer risk assessment tools and assess their ability to stratify screening populations according to (i) absolute risk of breast cancer and (ii) related outcomes of breast cancer risk (expected versus observed incidence of invasive breast cancer, with or without in situ disease and incidence of breast cancer). This review was undertaken as part of the Roadmap to Optimising Screening in Australia (ROSA) project [22] funded by the Australian Government Department of Health, and includes: (i) studies that compare tools generated from, or calibrated to, a different population to the one in which the tools were applied to, i.e., the validation population of interest, and (ii) studies comparing risk assessment tools calibrated or recalibrated to the validation population of interest. 

## 2. Methods

### 2.1. Study Registration

Our Patient, Intervention, Comparison, Outcomes (PICO) question is ‘For asymptomatic women aged ≥40 years, how accurately do different breast cancer risk assessment tools assign women to risk groups?’, where the term ‘risk assessment tool’ is used synonymously for risk prediction tool, prognostic model, risk prediction model, risk model, and breast cancer prediction model. The protocol for this systematic review was registered on the International Prospective Register of Systematic Reviews (PROSPERO) as part of a larger protocol exploring breast cancer risk assessment tools (CRD42020159232). We followed the requirements of the PRISMA 2020 guidelines for conducting and reporting of systematic reviews [23]. 

### 2.2. Eligibility Criteria

The current analysis was confined to articles comparing breast cancer risk assessment tools on the same study cohort; cohorts had to consist of asymptomatic women undergoing population mammographic screening. We excluded articles limited to cohorts of women undergoing diagnostic breast imaging, specific ethnic groups or women with high risk of breast cancer as these represent sub-groups of the screened population. We considered only external validation studies (so that the study cohort was different from that used to develop each tool being compared), We included randomised controlled trials, paired cohort studies or systematic reviews thereof. Due to the need for sufficient follow-up between risk assessment and cancer outcomes, we included prospective or retrospective cohort studies (based on timing of risk predictor data collection in relation to outcome occurrence). All other study designs (such as cross-sectional studies or case–control studies) were excluded.

We included risk assessment tools based on questionnaire data with or without genetic and/or breast density information, where estimated future risk was projected to a minimum of two years (in line with the most common screening interval of most population breast cancer screening programs). Tools designed to be calibrated to the target population prior to use were included if they were developed on a different population to the study cohort. Tools requiring any non-standardised input (e.g., subjective assessment by a clinician) were excluded.

We restricted our analysis to articles published from 2008, aiming to include studies likely to use more relevant imaging methods and more recent versions of risk assessment tools while not excluding relatively contemporary studies with longer periods of follow-up. Only English language peer-reviewed publications were included; conference abstracts, reviews, letters, editorials and comments were excluded.

The primary outcome was expected versus observed incidence of invasive breast cancer (with or without in situ disease). Secondary outcomes were breast cancer mortality, incidence for different types of breast cancer as defined by characteristics such as tumour subtype, grade, size, nodal involvement, and interval breast cancers (i.e., cancers diagnosed following a negative screen and before any consecutive screens). Articles that did not report expected versus observed (E/O) calibration outcomes according to risk groups determined by the risk assessment tool were excluded.

Results were excluded from the analysis if risk was projected beyond the period for which the tool was developed. Five-year risk was the primary outcome compared and reported; results for 10-year risk are included in Appendix A. 

We contacted corresponding authors when there was a lack of clarity around criteria for inclusion in our review, allowing two weeks for a response, after which we sent a reminder in addition to contacting other authors on the paper. If no response was received, the study was excluded. Extracted data is presented in Appendix A.

### 2.3. Information Sources and Search Strategy

An experienced systematic reviewer (VF) searched on 1 July 2021 for English-language reports published from 1 January 2008 to 29 June 2021 on the following databases: (i) Ovid Medline and Embase; (ii) The Cochrane Database of Systematic Reviews (CDSR) and (iii) PROSPERO. An updated search until 20 July 2021 was also performed for these databases. For Ovid databases, database-specific subject headings and text terms were combined for breast cancer, risk assessment and calibration terms (see Supplementary Methods). The CDSR was searched by combining “breast cancer” and “risk” text terms. Reference lists of relevant systematic reviews and full-text articles were also scanned for additional potentially relevant reports by two systematic reviewers (VF, DC). The search strategy is presented in Appendix A.

### 2.4. Selection Process

Titles and abstracts of the articles identified via the literature searches were screened against pre-specified inclusion criteria and split equally between two reviewers (VF, DC) with 20% assessed by both reviewers. The two reviewers independently assessed full-text articles of potential or unclear relevance for inclusion using a form with pre-specified selection criteria. Reviewers were not blinded to journal titles or study authors/institutions. Disagreements were resolved by discussion or adjudication by a third reviewer (SH). 

### 2.5. Data Collection 

Two independent reviewers (VF, DC) equally split the extraction of pre-determined study characteristics and results data from each included study and then reviewed the other’s extractions for accuracy. Disagreements were resolved by discussion or adjudication by a third reviewer (SH, LV or CN); experienced statisticians were consulted to advise upon or review article methodology or calculations (SE or CN).

The following information was extracted: first author, publication year, country, study design, setting, study start, participant inclusion/exclusion criteria, screening protocol, population characteristics, risk assessment tool information, follow-up duration, risk prediction interval, reported relevant outcomes, E/O estimates and 95% confidence intervals (CIs), observed rates (or if missing, the observed number of breast cancers and number of women in each risk category) and other relevant information (including methods used, factors potentially affecting risk of bias). If E/O ratios, their 95% CIs or data for observed rates were not reported, these were calculated by the systematic reviewers from available data or plots where possible (VF, DC, SE). Ninety-five percent CIs were calculated using the following formula: E/O × exp ^ (±1.96 × sqrt(1/O)) [23,24]. If there was insufficient data to perform calculations, authors were contacted and if attempts to obtain data were unsuccessful, the tool or study was excluded. In addition, where a tool version remained unclear after contacting authors and major updates to risk predictors had occurred between versions, the tool was excluded. It should be noted that risk predictors may be identified as risk factors, covariates, risk indicators, prognostic factors, determinants or independent variables [24].

We also identified high, moderate and low risk groups for each tool in each cohort. These groups were dependent on the number of quantiles the cohort of interest was divided into and whether they had the equivalent number of participants in each one. In general, when the cohort was divided in equal quartiles or deciles, we assumed the high-risk group corresponded to quartile 5 or deciles 9 and 10, the low-risk group corresponded to quartile 1 or deciles 1 and 2 while moderate-risk groups correspond to the remaining quantiles (quartiles 2–4 or deciles 3–8). 

### 2.6. Metrics for Evaluating Risk Assessment Tools and Statistical Analysis

Prior to analysis, risk assessment tool comparisons were grouped by comparator tool (which could be any version of that tool). Data was extracted into Microsoft Excel and then plotted for each tool, age range and predicted year of risk. 

We generated various data presentations and metrics to help evaluate and compare studies, as follows: A.Goodness of fit between expected (predicted) and observed outcomes:
1.Plotted ratios of expected versus observed cancers, by population percentile. The E/O ratio (in log10 scale) with 95% confidence intervals were plotted according to risk group assignment using the mid-point percentile of each risk group in the study population. This facilitated standardisation of comparisons between tools that had a different number of risk groups and/or assigned different proportions of women to each risk group.2.The total number of women in each study cohort in risk groups for which the E/O 95%CIs included unity. This helped indicate the proportion of each study cohort that was well-validated by the tool, noting that this is more likely for smaller studies (and therefore wider CIs).3.Calibration belt goodness-of-fit tests. We assessed goodness of fit between expected (predicted) probabilities of developing breast cancer and observed data using calibration belts [25] as applied in Li et al., 2021 [26], where a *p*-value <0.05 indicated miscalibration by the tool [25].B.Analysis of observed outcomes by risk group classification:
1.Observed cancer rates (number of breast cancers divided by the number of women per 10,000 for each risk category), by mid-point percentile of each risk group in the study population. This helped to standardise comparisons.2.Characterisation of the functional form (curve) of observed cancer incidence rates according to increasing risk group, classified as either: ‘increasing’ (observed rates consistently increasing across risk categories), ‘monotonic’ (i.e., increasing or remaining steady across groups) or ‘fluctuating’ (all other options).3.Assessment of whether highest-risk women could be distinguished from women at more moderate-risk. We compared the observed breast cancer rate corresponding to the mid-range risk groups (usually quintiles 2–4 or deciles 3–8) with the highest risk group (quintile 5 or deciles 9–10). *p*-values <0.05 indicated a statistically significant difference and, therefore, good allocation of women to the highest risk group. To ensure comparability of findings, if >25% of the study cohort was allocated to the highest risk groups, *p*-values were reported but not taken into consideration when drawing conclusions regarding a particular tool. Consequently, mid-range risk groups would be expected to include ≥50% of the study cohort.4.Assessment of whether lowest-risk women could be distinguished from women at more moderate-risk. As for (3 above), but for the lowest risk group (quintile 1 or deciles 1–2 or the equivalent sub-groups representing ≤25% of cohort), compared to the remainder (quintiles 2–4 or deciles 3–8, or equivalent sub-groups representing ≤50% of the cohort). To ensure comparability of findings, if >25% of the study cohort was allocated in the lowest risk groups, *p*-values were reported but not taken into consideration when drawing conclusions regarding a particular tool.

Plots and all statistical analyses were conducted using STATA (version 17, Stata Corporation, College Station, TX, USA). 

### 2.7. Risk of Bias Assessment 

Two independent reviewers (DC, VF) assessed the risk of bias for each included study. Differences were resolved by consensus or adjudication from a third reviewer (JS). Risk of bias was assessed using the ‘Prediction model Risk Of Bias ASsessment Tool’ (PROBAST), specifically designed to assess the Risk of Bias for, and the applicability of, diagnostic and prognostic prediction model studies [24]. PROBAST is organised into four domains; (i) participants (assessing suitable data sources or study designs and appropriate inclusions or exclusions), (ii) predictors (assessing predictor definition and measurements, knowledge of outcome influencing predictor assessment and whether the tool is used as designed if predictors are missing at time of validation), (iii) outcome (assessing methods used to classify participants with or without outcome, pre-specified/standard definition of outcome used, predictor exclusion from outcome definition, similar definition and determination of outcome for all participants, knowledge of predictor influencing outcome assessment, time interval between predictor assessment and outcome determination), and (iv) analysis (assessing reasonable number of participants with outcome, handling of continuous and categorical predictors, enrolled participant inclusion in analysis, handling of participants with missing data, handling of data complexities, evaluation of relevant tool performance measures). Each domain contains signalling questions to facilitate a structured judgement of risk of bias; the overall rating for a domain can be classified as either “low”, “high” or “unclear” risk of bias. Each study is also allocated an overall risk of bias rating: “low”, if no relevant shortcomings were identified in the risk of bias assessment; “high”, if at least one domain was assessed as high risk of bias and “unclear” if risk of bias was assessed as unclear for at least one domain (and no other domains assessed as high risk of bias).

For each study, a separate risk of bias assessment was conducted for each distinct risk assessment tool validated, for each individual outcome and each cohort included [24]. Outcomes with multiple time points (e.g., 5- and 10-year risk predictions) were assessed separately because ratings for signalling questions on appropriate time interval between predictor assessment and outcome determination, and reasonable number of participants with outcome, could differ. As such, it was possible for a single study to have multiple overall risk of bias assessments. 

Rulings were developed where necessary to account for judgements that required topic-specific knowledge or statistical expertise. These rulings were initially trialled independently over several studies by the same two reviewers (DC, VF) with third reviewer input from a senior researcher (JS) where required. It was decided a priori that: (i) risk of bias domains that contained signalling items relating only to model development would be omitted as the primary interest of this systematic review was risk assessment tool validation and (ii) the applicability of a study would not be formally assessed by the PROBAST tool; instead, concerns would be highlighted where necessary in the discussion. We sought statistical advice to develop rulings for items in the analysis domain as suggested by PROBAST. When assessing the reasonable number of participants with outcome PROBAST recommends that validation studies should include at least 100 participants with outcomes. After consulting statistical experts (SE, DO’C), it was decided a priori that a study would qualify for a low risk of bias rating if this was the case for every risk category. Where data for observed incidence of breast cancer per risk category was provided by authors or calculated by reviewers from calibration figures, this was used to inform our ratings. Otherwise, risk of bias was appraised based on the information reported in the article and included references. For the handling of missing data, based on methodological advice (QL), it was decided a priori that a study performing multiple imputation would qualify for low risk only if <50% of values were originally missing (and thus imputed) for a predictor and the missing data were missing at random [27,28].

## 3. Results

### 3.1. Selection of Articles and Summary Characteristics

Figure 1 summarises the search process conducted. The search strategy identified a total of 5114 records of which 3405 remained after duplicates were removed. Of these, 3324 records were excluded based on title and abstract review. Full texts or records of 91 potentially relevant reports were assessed according to the eligibility criteria. This included 10 additional articles identified from citation searching of full text articles and 1 potentially eligible article from the update search conducted on the 20 July 2021. A total of 78 reports were found not to be relevant and therefore excluded. We contacted authors to confirm the eligibility of two tools for one validation cohort [29]. Common reasons for exclusion included ineligible study design, ineligible population and E/O not reported by risk category. Further details on reasons for exclusions (studies and tools) and information regarding authors contacted are details in a Appendix A.

The remaining 13 articles included in this review examined the prediction of breast cancer across 15 cohort studies applying 11 distinct breast cancer risk assessment tools of different versions. Summary characteristics of included articles are presented in Table 1. All studies were prospective in design apart from one retrospective study [30]. Ten of the 13 articles were from North America and Europe and compared more than two risk assessment tools based on a 5-year risk prediction interval. Only two articles presented findings for 5- and 10-year tool-determined risk [31,32]. 

The tools assessed included data from questionnaires, with or without information on mammographic breast density and PRS. The number of risk predictors varied between tools, from as few as five (e.g., Chen version 1) [41] to as many as 13 (e.g., Tyrer-Cuzick version 8.0b [31] although it should be noted that some studies did not have data for all the risk predictors specified by the tool they assessed. Risk predictors considered in each tool by each study are presented in Appendix A. The Breast Cancer Risk Assessment Tool (BCRAT; also known as the Gail model), was the most frequently assessed tool in publications assessed (9 of 13 articles). This was followed by the Tyrer-Cuzick tool (also known as the IBIS risk assessment tool) in 5 articles, and BRCAPRO and iCARE-Lit in 3 articles each. 

Two articles evaluated the effect of adding breast density data: McCarthy et al. [32] compared 5-year risk using Tyrer-Cuzick version 7 versus version 8.0b which had breast density incorporated within the tool and Brentnall et al. [38] assessed 10-year risk using Tyrer-Cuzick version 7.0 with and without breast density data. In two more articles, tools with integrated breast density data (Chen version 1; Tyrer-Cuzick version 8.0b) were compared to other tools; in Choudhury et al., 2020 [35] Tyrer-Cuzick version 8.0b was compared to iCARE tool variants, and in Arrospide et al. [41] Chen version 1 was compared to BCRAT version 1.

Only one study assessed the effect of PRS data on existing tools; in Hurson et al. [29] a 313-variant polygenic score was added to two iCARE risk assessment tools (iCARE-Lit and iCARE-BPC3). 

Evidence was available to compare tools in terms of risk of invasive breast cancer, however, evidence was sparse for in situ breast cancer incidence, while no data was available on breast cancer incidence according to prognostic indicators (e.g., tumour subtype, grade, size, nodal). Therefore, these outcomes were not able to be assessed.

### 3.2. Goodness-of-Fit

Absolute risk calibration is shown for various tools and tool comparisons (along with observed rates of incident breast cancer) in Figure 2A–C and Appendix A. In terms of goodness-of-fit between estimated and observed outcomes, no risk assessment tool was identified as being consistently well-calibrated in multiple studies. As can be observed from Table 2, many tools showed good calibration in some but not all studies: namely AABCS [32,40], BCRAT version 3 [30,35,36], BCRAT version 4 [31,33,34], Tyrer-Cuzick version 8.0b [31,33,34,35], iCARE-Lit and iCARE-BPC3 [29,35], and BRCAPRO version 2.1 [31,34]. In contrast, some tools did not demonstrate good calibration across studies; examples include BCRAT version 2 [32,40] and Tyrer-Cuzick version 7 [31,34]. There were other tools that were applied in single cohorts within this review, and thus could only be assessed in only one population and one setting. Of these, six showed a good fit (BCRAT version 1 [41]; Chen version 1 [41]; BCRmod [36]; BCRmod recalibrated [36]; KREA for women over 50 years [37]; KRKR [37]) and five showed evidence of miscalibration (i.e., *p* < 0.05) (BOADICEA [31]; ER- [39]; ER+ [39]; KREA for women under 50 years of age [37]; original Korean tool [40]; updated Korean tool [40]).

Combining breast density data with a tool score [38] or integrating breast density within a tool generating a new tool version [Tyrer-Cuzick version 7 vs. Tyrer-Cuzick version 8.0b [34]) did not improve the goodness of fit of the tool, with evidence of miscalibration in both cases. 

Addition of PRS data, in the single study that evaluated a specific score [29], did not improve the goodness-of-fit of neither iCARE-Lit nor iCARE-BPC3, as assessed on different cohorts ([UK Biobank; Women’s Genome Health Study (US)] (Appendix A). For these evaluations, there was evidence of miscalibration before and after addition of PRS information (*p* < 0.05). 

No change was observed in the calibration of most tools (BCRAT version 2; BCRAT version 4; BRCAPRO version 2.1; BOADICEA) for longer-term risks, with evidence of miscalibration for both 5- and 10-year risk. The only exception was the AABCS tool, for which the goodness-of-fit improved for 10-year risk [32]. 

### 3.3. Observed Cancer Incidence by Risk Group

The majority of tools, with a few exceptions (Chen version 1 [41], ER- [39], the KREA and KRKR Korean tools [37] and the original Korean model [40]), were able to identify the broad group of women with the highest risk of breast cancer. This group always corresponded to the highest observed rates of incident breast cancer, indicating that most tools are effective in identifying women in the highest risk category in one setting (BCRAT version 1 [41]; BCRAT version 4 [34]; BCRmod and BCRmod recalibrated [36]; ER+ [39], Tyrer-Cuzick version 7 [34], updated Korean [40]) and across different settings (AABCS and BCRAT version 2 [32,40]; BCRAT v3 [35,36]; iCARE-Lit and iCARE-BPC3 [29,35]; BRCAPRO [30,34] and Tyrer-Cuzick version 8.0b [34,35]).

Some tools could consistently stratify women in the lowest categories of breast cancer risk across different settings; namely Tyrer-Cuzick version 8.0b [34,35], BCRAT version 3 [35,36], BRCAPRO [30,34], iCARE-Lit and iCARE-BPC3 [29,35]. Although additional tools could distinguish women in the lowest category of risk in a single setting (e.g., BCRAT version 4, Tyrer-Cuzick version 7, ER+, KREA, KRKR), there was not enough evidence to ascertain their performance across different settings. 

The contribution of PRS to improving risk tool accuracy varied between tools and sub-groups in Hurson et al. [29]. For example, PRS improved the consistency of the graded association between risk groups and observed rates for the iCARE-Lit tool applied to a UK cohort of women aged under 50 years but did not improve the trend for women in that cohort aged 50 years or older, nor for a US cohort aged 50–74 years. Another iCARE tool variant of (iCARE-BPC3) worsened the graded association between risk groups and observe cancer rates for women aged 50 years or older in a UK cohort. 

The addition of mammographic density appeared to improve some tools slightly for some risk groups. For example, the Tyrer-Cuzick tool reported in McCarthy et al. [34] improved differentiation for the higher-risk groups but worsened the graded association in lower-risk groups (Figure 2A), and Tyrer-Cuzick applied in Brentnall et al. [38] did not discernibly improve the association (Appendix A).

There was limited evidence to evaluate the effect of a longer risk-prediction interval on observed cancer incidence. The AABCS tool appeared to better differentiate lower and higher-risk groups at 10 years than 5 years, [32,40] and the BCRAT version 2 tool was more clearly graded with longer-term cancer incidence (Figure 2B; Appendix A) [32]. It was not possible to evaluate the results from the risk assessment tools reported by Terry et al. [31] due to the uneven distribution of the cohort among the five risk groups reported.

### 3.4. Risk of Bias Assessment

Risk of bias assessments were undertaken for each of the tools evaluated in each study. The overall risk of bias rating for all 47 risk of bias assessments undertaken was high (Table 3). The overall risk of bias for the participants domain was low for 75% of assessments. For the predictor domain, the overall rating was low for 36% of assessments, high for 36% and unclear for 28%; for the outcome domain, 66% of assessments were rated as unclear and 34% at high risk while the overall rating for the analysis domain was high risk of bias for all 47 assessments. Detailed findings listed per risk of bias domain are provided in the Appendix A. 

Common factors that contributed to unclear or high risk of bias ratings included: handling of missing predictors at the time of validation when a tool did not allow for an unknown or missing option; specification of standard measurement of predictors at baseline; minimal reporting of predictor assessments blind to outcome; limited information provided around methods used to determine outcomes; omission of standard outcome definitions and standardised follow-up protocol and lack of clarity on the number of women who had full follow-up for the time interval between predictor assessment and outcome determination. Furthermore, the analysis domain rated poorly with all tools examining 5-year risk having <100 events across risk categories, although this was achieved for most tools assessing 10-year risk. Additionally, often no direct reference was made to baseline questionnaires preventing a clear assessment of the handling of continuous and categorical predictors (i.e., if data transformation was required between collection vs. input) unless stated in the text.

Tools tended to rate poorly for methods regarding handling of missing data. The main reasons for poor ratings included inappropriate assumptions, omitting predictors with missing data in general or for a particular predictor, and imputation of predictors with >50% missing participant data. 

## 4. Discussion

### 4.1. Summary of Main Results

This systematic review of studies comparing multiple breast cancer risk assessment tools within general populations examined several metrics to evaluate risk assessment tools, namely: the ratio of the expected over observed number of breast cancer cases; evidence of miscalibration; the proportion of study group where E/O and 95%CI includes unity, and how these related to the observed cancer incidence rates across assigned risk groups. We found that no tool was consistently well-calibrated across multiple studies, and breast density or polygenic risk scores did not improve calibration. While most tools identified a risk group with higher rates of observed cancers, few tools identified lower-risk groups across different settings. 

We did not apply a single metric to compare tools because the interpretation and value of each metric depends on how the risk assessment tool might be used. Where risk assessment tools are being used to advise an individual woman about her estimated breast cancer risk, specified as, for example, her 5-year or lifetime risk, the tool should have demonstrated very good calibration of E/O rates within her population to ensure a sufficiently accurate estimate. Communication of this information is also important, as these estimates are often misinterpreted as individual level risk so that, for example, an estimated 3% five-year risk is interpreted as the individual woman having a 3% risk of breast cancer in the next five years, when instead it indicates that 3% of women in the risk group to which she belongs would be expected to have a breast cancer diagnosed in the next five years [42].

The individual-level risk estimates generated by risk tools are also used in clinical practice to advise and manage women according to a risk group assignment based on their estimated risk of breast cancer, without necessarily reporting the estimated individual breast cancer risk for each woman. For example, the Royal Australian College of General Practitioners (RACGP) guidelines define women at ‘moderately higher’ risk as those with a 1.5 to 3 times higher than average risk, and women at ‘potentially high’ risk as more than 3 times the average population risk and recommend management based on these risk categories such as screening frequency and/or referral to specific breast imaging surveillance tests, or referral to specialist high-risk services [43]. These tools generally rely on individual risk estimation as the basis for risk group allocation. For example, the iPrevent tool draws on either the IBIS or BOADECIA risk tool depending on an assessment of initial factors such as family history, then assigns the individual to a risk group following the RACGP guidelines. Each woman’s risk relative to the average is defined by the ratio of her estimated residual lifetime risk (to age 80) and the average residual lifetime population risk for women of her age. In a validation study of over 15,000 Australian women, iPrevent demonstrated good calibration for women under 50 years (E/O: 1.04; 95% CI = 0.93 to 1.16) but poor calibration for women aged 50 years and older (E/O: 1.24; 95% CI = 1.11 to 1.39), largely due to overestimation of risk in the highest study group decile [44]. These findings are concerning in terms of providing accurate risk estimation to individual women however, as noted by Phillips et al., “the extent of overestimation is unlikely to be of clinical importance because the actual 10-year [breast cancer] risks for these women substantially exceed thresholds for intensified screening and medical prevention (and for mutation carriers, risk-reducing mastectomy). Therefore, the overestimation would be unlikely to lead to an inappropriate change in their clinical management.” 

The issues mentioned above have potential consequences for how risk assessment tools should be evaluated in relation to risk-based population breast screening While GPs and specialists are (theoretically) able to refer an unlimited number of patients to services to which they are eligible, resource-constrained population risk-based screening programs would benefit from directing screening protocols for higher-risk (or lower-risk) clients to a priori proportions of the screening population. This could mean, for example, that a screening program would provide supplemental or alternative imaging tests to 10% of women deemed to be most likely to benefit from that imaging, based on their short-term breast cancer risk and the expected accuracy of their routine screening test (indicated by, for example, observed interval cancer rates). For this purpose, it should be sufficient to confirm that a risk tool can identify the 10% of screening clients for whom outcomes (observed rates of breast cancer and interval cancers) under the current approach to screening are significantly higher compared to clients with average outcomes in the screened population, even if that tool is not well calibrated in terms of expected and observed rates; this risk stratification could then be used to trial alternative approaches to screening.

This is an important consideration because requiring good E/O calibration of risk assessment tools across the risk spectrum is a difficult standard to reach. For example, a recent evaluation of six established risk models (IBIS, BOADICEA, BRCAPRO, BRCAPRO-BCRAT, BCRAT, and iCARE-lit) in over 52,000 Australian women concluded that only one model (BOADICEA) calibrated well across the spectrum of 15-year risk [26]. 

Even where good E/O calibration is achieved, this does not necessarily mean that observed rates are ranked well or that calibration is good across the risk spectrum. For example, in the study by McCarthy and colleagues [34], despite BRCAPRO exhibiting goodness-of-fit for the cohort, the observed rates fluctuated for women in the middle deciles, and the assessment of the KRKR tool by Jee et al. [37] on women aged 50 years or older demonstrated good calibration overall but was well-validated for only 30% of the risk groups (of note, this metric is more stringent for studies with a larger number of risk groups such as Jee et al., which had ten). Conversely, models with evidence of miscalibration can demonstrate good differentiation of a higher-risk group. For example, in the study by Hüsing et. al., although BCRmod recalibrated showed evidence of miscalibration to the study population, higher-risk groups (deciles 9–10) were well differentiated [36].

Overall, despite differences between risk assessment tools and study cohorts, most risk tools were able to identify a group of women with the highest risk of breast cancer, with only a few exceptions (Chen v1, ER-, KREA, KRKR and the original Korean model). For lower-risk women, some tools assessed consistently stratified women in the lowest categories of breast cancer risk across different settings (e.g., Tyrer-Cuzick version 8.0b; BRCAPRO version 2.1; iCARE tools). In the case of BCRAT, this depended on the version used; i.e., BCRAT version 3 was found to be consistent in distinguishing women in the lowest risk group whereas the same was not observed for versions 2 and 1. Of note, for some tools it was not possible to assess this feature across different settings as there was only one relevant study included (e.g., BCRmod [36], KREA and KRKR [37], ER+ tool [39]).

The BCRAT tool was the most evaluated risk assessment tool in the included articles, followed by the Tyrer-Cuzick tool, with increasing evaluation of iCARE tools in more recent publications. The number of risk factors considered by the different tools varied considerably. This is an important consideration for policy-makers and health services when selecting the most suitable tool for a specific application, as the number of predictors and the level of detail required for each one can be an impost for women and requires substantial resources to ensure complete and accurate risk information is provided and recorded. 

The number of risk groups varied greatly between studies (4–10 groups). Reporting results for more groups provides more detail on how the tool performs as a graded association with increasing risk, which is informative for population-level applications where the availability of resources might be limited. For example, isolating smaller groups of women with very high risk may be more feasible for targeting more costly options (such as MRI) to higher-risk women as part of population breast screening.

We found that mammographic breast density has not been shown to improve the accuracy of breast cancer risk assessment tools based on self-reported information collected from questionnaires. We did not review evidence on the accuracy of breast density alone as a risk assessment tool, with an equivalent assessment of whether other risk predictors improved the accuracy of breast density as a risk assessment tool. However, this is a very active research area, and ongoing review of high-quality evidence is warranted. 

Similarly, we found that the addition of a PRS score did not improve accuracy when added to self-reported information within the tools assessed, although this finding was based on a single study [29]. We did not review evidence on the accuracy of PRS alone as a risk assessment tool. 

### 4.2. Comparison with other Published Work

A number of other systematic reviews have been published previously in this field [45,46,47,48]. These aimed to provide an overview of published risk assessment tools, basing their assessments on (i) calibration performance using the E/O ratio and (ii) discriminatory accuracy using the area under the receiver operating curve (AUC) and/or concordance statistic (C-statistic). In this review we focused on studies that assessed more than one risk assessment tool on one or more populations, how those tools compared to each other and what overall observations could be drawn by assessing these studies collectively. For this purpose, the AUC and C-statistic are not considered the appropriate metrics for assessing discrimination as they measure the ability of a tool to determine which women are at higher or lower risk of breast cancer than average, but not whether women within a study population have been stratified according to their level of risk, which is critical when evaluating these tools for the purpose of population-based risk-based screening. We recommend the use of observed rates of incident breast cancer according to tool-determined risk groups as it provides a better quantitative assessment of discrimination for this purpose, informing consideration of interventions that might target women at different thresholds of risk across the risk spectrum. 

### 4.3. Applicability and Model Performance

We observed that tools that were recalibrated to the risk profiles of the population in which they were applied demonstrated an improvement in fit, as exemplified in the study by Chay and colleagues [32], which compared BCRAT to its Asian-American variant. This improvement in a risk assessment tool highlights the importance of making such adjustments when considering the application of any risk tool, especially on specific populations. Tools are usually developed using breast cancer incidence rates and risk factor data collected from one population and then applied to a different population without adjusting these parameters. This can lead to poorer model performance as the distribution of risk factors and breast cancer incidence can vary across populations. We need, however, to distinguish between recalibration and ‘pre-calibration’ as exemplified by the iCARE-based tools which uniquely incorporated calibration to population-based age-specific disease incidence rates before they were used [35]. As can be seen from Table 2 and associated graphs, these tools generally performed very well. They fell within the scope of this systematic review as they met the review’s criterion of a tool calibrated to the study validation population of interest. This approach in the use of risk-prediction tools seems sensible given that population-based age-specific disease incidence is usually available and, as reinforced by this review, tools without calibration perform very differently in different settings.

Assessment of the studies included, revealed opportunities to improve standardisation of risk tool evaluations. Not all studies cited the specific version of the tool and package used. When these details were not provided, it was difficult for reviewers to deduce this information even if predictors were listed. For example, one study [36] provided a link to the BCRAT tool on the National Cancer Institute (NCI) website and a second study [35] using the same tool, also included the date and year accessed. However, the NCI provides the latest tool versions without detailed history of previous versions and updates; therefore, the version of the tool used by these studies at the time they were conducted had to be deduced. For studies that cited tool versions, these were often determined by the software used; e.g., BCRAT can be run on SAS Macro or R and these packages have their own tool-version numbers. For some models, the software was accessible through different sources. For example, BRCAPRO is accessible via the BayesMendel R package or within the CancerGene software program which now uses the code from BayesMendel. In the case of studies using the latter, even when the CancerGene software version was cited there was insufficient information available from the CancerGene website to deduce which version of the BayesMendel R package was used by that software. For full transparency it is recommended that authors provide the specific version of the risk assessment tools used including the software package, all predictors offered by that version and used in the study being reported. 

### 4.4. Risk of Bias and Quality of the Evidence

Critical assessment of studies in terms of risk of bias is required to provide a comprehensive evaluation. We used the recently published PROBAST tool, specifically designed to thoroughly assess the risk of bias in relation to risk assessment tool studies. Only one previous systematic review identified from our searches had included a risk of bias assessment, although a tool for evaluating modelling studies was used instead of PROBAST [42]. All tools we evaluated across studies received an overall rating of ‘high risk of bias’. Although this was driven mainly by rulings for the domain of analysis, there was also an evident lack of clarity in the reporting of key details contributing to ratings of ‘unclear risk of bias’ for 28–66% of tools for the predictor and outcomes domains. 

One of the main areas of concern is the domain of predictors with respect to the collection and completeness of data on risk predictors, and the statistical methods used to deal with any data issues. One method that studies used to deal with missing predictors at the time of validation was multiple imputation (29, 35). Although this is a common method to deal with missing data, the reference dataset is simulated and thus possibly less reliable. This also limits our understanding of how missing data would be addressed at an individual level if the tool were utilised as part of health service provision. In other studies, researchers sometimes stated that missing data was handled according to the specifications of each software application (e.g., McCarthy et al. [34], Jantzen et al. [33]), however it was not always clear whether a predictor value was then classed as missing or whether the predictor was omitted from the tool (e.g., BRCAPRO version 2.1–3, Tyrer-Cuzick version 8.0b and BOADICEA v3 in Terry et al. [31]). In other cases, the approach to handling missing data was not reported. For example, Brentnall et al. [38] applied version 7.02 of the Tyrer-Cuzick model (developed in the UK) on a US cohort; this version included prior use of hormone replacement therapy (HRT) (yes/no) as a predictor without an option of selecting ‘unknown’ or ‘missing’ if these data were unavailable. The authors did not report any information regarding the collection of HRT data or how missing data was handled. Overall, it is not possible to evaluate the precise effect of missing predictor values on risk estimates unless provision of a ‘missing’ option has been made by tool providers, which may indeed be more reflective of actual use of tools in practice as sometimes information on predictors cannot be recalled. We recommend that future studies consider including information on how missing data are managed, as this would improve comparability between studies and help recognise the challenges of applying risk assessment tools to different settings and study populations. Overall, factors identified by our risk of bias analysis could potentially explain some of the observed differences in tool performance in different settings described throughout this review. 

We also recommend more standardised and transparent reporting of risk assessment tools, using the ‘Transparent Reporting of a multivariable prediction model for Individual Prognosis or Diagnosis’ (TRIPOD) statement published in 2015 [49]. TRIPOD provides a 22-item checklist considered to be key for transparent reporting of risk assessment tool studies. The statement was created to increase the level of reporting standards as prior studies performing external validation of risk assessment tools were found to commonly lack clarity in reporting and tended not to present important details needed to understand how the tool might be applied or whether results reflected true performance of the tool [50]. This was reflected in a systematic review examining the methodological conduct and reporting of external validation studies for risk assessment tools that found that of 45 articles published in 2010, 16% did not report the number of outcome events when validating tools, 54% did not acknowledge missing data, and frequently, it was unclear as to whether the authors had applied a complete or an abridged version of the tool [50]. For our analysis, four studies [30,32,40,41] were published prior to the TRIPOD statement, however no studies published after 2015 refer to the TRIPOD statement or checklist. 

### 4.5. Limitations

This systematic review has certain limitations. A number of studies that compared different risk assessment tools on the same population were not included due to the focus of this review to compare risk assessment tools generated from, or calibrated to, a different population to the study validation population of interest or tools specifically calibrated to the study population of interest. However, focusing on the selected studies in this review enabled a fairer comparison between tools and improvement in the quality of the evidence. Secondly, despite meeting the criteria for inclusion, some studies had to be excluded due to some required data being unavailable for full assessment. Nonetheless, efforts were made to contact the authors. Additionally, for studies which did not provide the number of women in each risk category, the calculated estimates may be inaccurate if numbers are distributed unequally between risk categories, as the number of women per category was estimated by dividing the numbers of participants equally among categories. 

This review did not compare tools in terms of interval cancers (i.e., cancers diagnosed following a negative population screening test), breast cancer mortality, nor incidence of breast cancer defined by different tumour characteristics (e.g., sub-type, size, grade, nodal involvement). We did initially seek to assess these outcomes as this evidence is likely to be of interest for some applications, such as consideration of risk-based screening protocols, however insufficient evidence was available to make these comparisons between tools. 

Finally, one of the methods used to assess risk assessment tools was based on E/O point estimates and their 95%CI including unity (E/O = 1). Studies where tools were applied on small cohorts of women have wider CIs and therefore be more likely to include this value compared to larger studies which have narrower CIs. Additionally, we characterised the functional form of observed cancer rates according to risk groups based on point estimates reported without uncertainty estimates (e.g., CIs). However, while we acknowledge these metrics have minor limitations, these were only two of the metrics employed; when evaluated collectively all metrics analysed provide sufficient information to enable a fair and balanced assessment of risk assessment tools.

## 5. Conclusions

This systematic review identified various questionnaire-based tools (sometimes incorporating mammographic density or genetic information) that are effective in assigning women to risk groups for incident breast cancer, for various metrics of tool performance. The most appropriate metrics to consider depend on how the risk tool is to be applied. While good calibration between expected and observed rates is essential for individual-level estimated breast cancer risk described as a rate over a specified period, tools demonstrating good differentiation of observed breast cancer incidence rates are potentially suitable for triaging women to population-level risk-based interventions such as risk-based breast cancer screening, even if they are not well calibrated in terms of expected versus observed outcomes across the risk spectrum. Current trials such as MyPeBS [51] and WISDOM [52] are allocating women to risk-based screening protocols based on their predicted risk of breast cancer as estimated by combining genetic information with scores from risk assessment tools which incorporate mammographic density (Tyrer-Cuzick and Breast Cancer Surveillance Consortium (BCSC) risk tools for MyPeBS [53]; BCSC risk tool for WISDOM [54]). Results from these studies will provide valuable information on the clinical utility of these detailed and resource-intensive risk assessment tools; in parallel, work is required to understand the relative utility of more parsimonious tools that may achieve similar outcomes while markedly reducing the impost of risk assessment on women and health services.

## Figures and Tables

**Figure 1 cancers-15-01124-f001:**
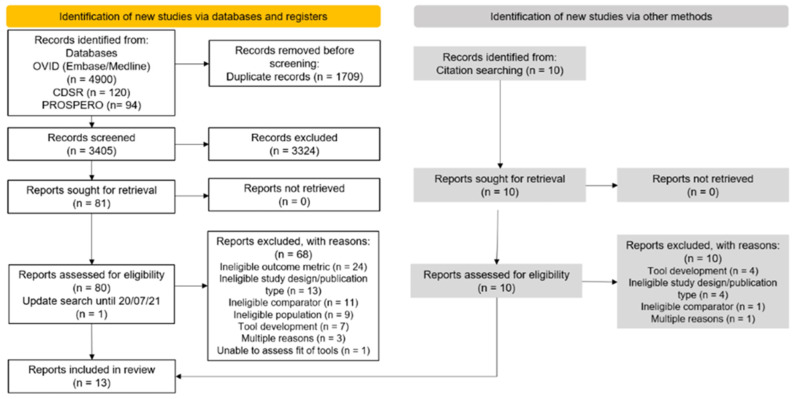
Flow diagram based on the PRISMA 2020 flow chart summarising the article screening process.

**Figure 2 cancers-15-01124-f002:**
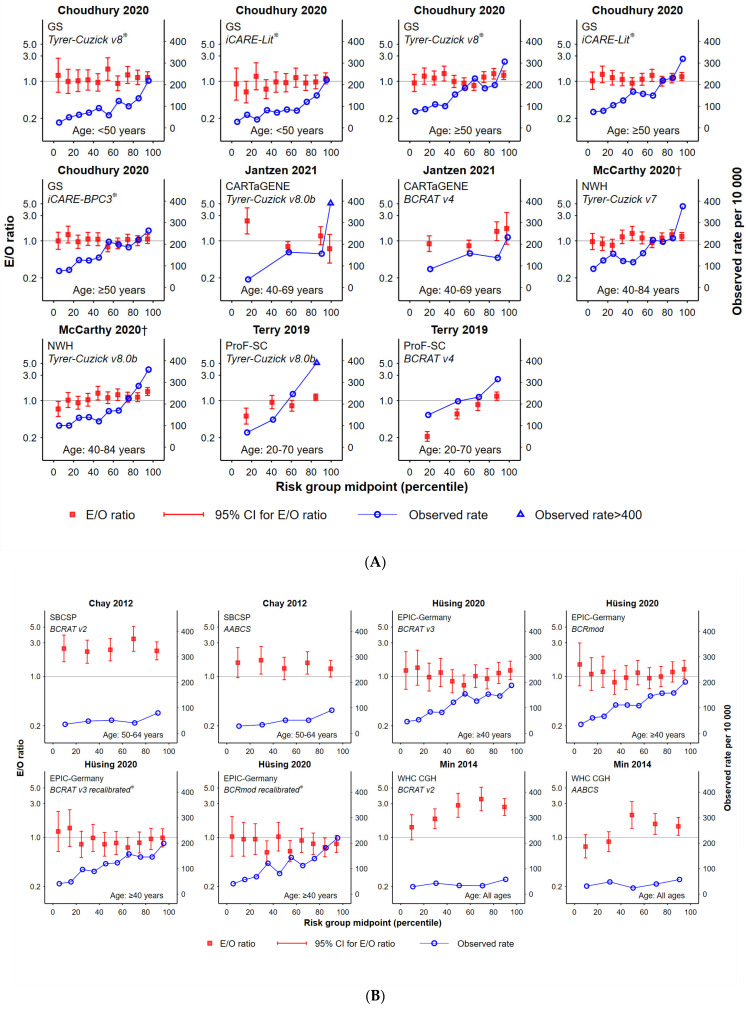
Absolute risk calibration and observed rate of incident breast cancer by 5-year risk. The three groups shown are: (**A**) Tyrer-Cuzick vs. BCRAT or other tool comparisons; (**B**) BCRAT vs. BCRAT modifications; (**C**) BCRAT vs. other risk assessment tools. Plots are then presented according to first author name. (The number of data points in each graph is determined by the number of risk groups that were reported in each study. To assist with comparison of studies, the x-axis shows the percentile distribution of groups being reported, with data points shown for the mid-points of each group. Red squares show the ‘expected over observed’ ratio for each risk group (with 95% confidence intervals shown), indicating calibration between expected and observed cancers at a risk group level. Blue circles show the corresponding observed rate of breast cancers within the study group, indicating the gradient of rates across the risk groups (expected to increase from left to right in accordance with increases in estimated breast cancer risk). Italic font indicates the risk tool being assessed, with the study cohort abbreviation also shown). * tools were calibrated to local population.

**Table 1 cancers-15-01124-t001:** Summary of the main characteristics of the included articles.

		Population									Outcome
Study ID	Country	Cohort	Age Range (Median), y	N	Study Start	Screening	Tool Comparisons	FU	Calibrated to Population?	Breast Cancer	Risk Interval (y)
Jantzen 2021 [33]	Canada	CARTaGENE	40–69 (53.1)	10,200	2009–2010	2 yearly, 50–69 y	TC v8.0b vs. BCRAT v4 ^a^	5	No	Invasive	5
Hurson 2021 [29]	UK	UK Biobank	Age subgroups <50 years: 40–49 at DNA collection; (46) ≥50 years: 50–72 at DNA collection; (61)	<50 years: 36,005 ≥50 years: 134,920	2006	NR	iCARE-BPC3 vs. iCARE-BPC3 + PRS iCARE-Lit vs. iCARE-Lit + PRS	4	Yes Yes	Invasive or DCIS	5
	USA	WGHS ^b^	50–74 at DNA collection; (56)	17,001	2000	NR	iCARE-Lit vs. iCARE-Lit + PRS	21 ^d^	Yes Yes	Invasive or DCIS	5
McCarthy 2020 [34]	USA	Newton-Wellesley Hospital	40–84; (53.9) ^d^	35,921	2007–2009	NR	TC v7 vs. TC v8.0b BCRAT v4 ^a^ vs. BRCAPRO v2.1–4	6.7 ^d^	No Yes	Invasive	6
Choudhury 2020 [35]	UK	Generations Study	Age subgroups <50 years: 35–49; (42) ≥50 years: 50–74; (58)	<50 years: 28,232 ≥50 years: 36,642	2003–2012	NR	TC v8 vs. iCARE-Lit, TC v8 vs. iCARE-BPC3, BCRATv3 vs. iCARE-Lit, iCARE-BPC3 vs. iCARE-Lit aRAT ^c^	5	Yes Yes Yes	Invasive	5
	USA	PLCO	50–75; (61)	48,279	1993–2001	NR	BCRAT v3 ^a^ vs. iCARE-Lit aRAT ^c^	5	Yes	Invasive	5
Hüsing 2020 [36]	Germany	EPIC-Germany	20–70; (40+: median 52.6)	22,098	1994–1998	NR	BCRAT v3 ^a^ vs. BCRmod BCRAT v3 ^a^ recalibrated vs. BCRmod recalibrated	11.8	No Yes	Invasive	5
Jee 2020 [37]	Republic of Korea	KCPS-II Biobank	Age subgroups <50 years: 21–49; (38) ≥50 years: 50–80; (58)	<50 years: 57,439 ≥50 years: 19,776	2004–2013	2-yearly, ≥40 years	KREA vs. KRKR (iCARE-Lit—based tools) aRAT ^c^	8.6	Yes	Invasive	5
Terry 2019 [31]	USA, Canada, Australia	ProF-SC	20–70; (NR)	15,732	1992–2011	NR	BCRAT v4 ^a^ vs. BRCAPRO v2.1–3; TC v8.0b vs. BCRAT v4 ^a^; BOADICEA v3 vs. BRCAPRO v2.1–3; BOADICEA v3 vs. BCRAT v4 ^a^	11.1	No No No No	Invasive	5, 10
Brentnall 2018 [38]	USA	Kaiser Permanente Washington BCSC	40–75; (50) (general population: ≥50 y; high risk: ≥40 y	132,139	1996–2013	Annually; 50–75 y; high-risk women 40–49 y ^e^	TC v7.02 vs. TC v7.02 + breast density	5.2	No	Invasive	10
Li 2018 [39]	USA	WHI	50–79; (63.2) ^d^	82,319	1993–1998	NR	ER- vs. ER+ aRAT ^c^	8.2 ^d^	No	Invasive	5
Min 2014 [40]	Republic of Korea	Women’s Healthcare Center of Cheil General Hospital, Seoul	<29 to ≥60; (NR)	40,229	1999–2004	NR	BCRAT v2 ^a^ vs. AABCS Original Korean tool vs. Updated Korean tool	NR	No Yes	Invasive	5
Powell 2014 [30]	USA	MWS	<40 to ≥80; (NR)	12,843	2003–2007	NR	BCRAT v2 or v3 ^a^ vs. BRCAPRO v(NR) aRAT ^c^	NR	Yes	Invasive	5
Arrospide 2013 [41]	Spain	Screening in Sabadell-Cerdanyola (EDBC-SC) area in Catalonia	50–69; (57.0) ^d^	13,760	1995–1998	2-yearly; 50–69 y	BCRAT v1 ^a,f^ vs. Chen v1	13.3	Yes	Invasive	5 ^g^
Chay 2012 [32]	Singapore	SBCSP	50–64;^h^ (NR)	28,104 ^i^	1994–1997 ^k^	Single 2-view mammogram, 50–64 y	BCRAT v2 ^a^ vs. AABCS	NR	No	Invasive	5, 10

^a^ Different versions of the BCRAT are labelled according to the SAS Macro version; ^b^ Following communication with the authors, iCare-BPC3 was excluded as part of the WGHS cohort was used for the development of this tool, ^c^ aRAT = Additional risk assessment tool. Additional tools were available for some studies but were excluded as they did not meet the criteria for inclusion in data synthesis, see supplementary methods for details); ^d^ Mean; ^e^ 62% of women aged <50 years at entry were low risk for breast cancer; ^f^ The study did not include DCIS in the outcome and women with DCIS were considered at risk of invasive breast cancer; ^g^ only 5-year risk data was extracted; ^h^ some women were older than 64 years based on screening time; ^i^ numbers or ages are as cited in text or tables; cannot verify accuracy due to different numbers or ages cited between the original trial and other reports; ^k^ organised national breast screening in Singapore was introduced in 2002. Abbreviations: AABCS = Asian American Breast Cancer Study; BCRAT = Breast cancer risk assessment tool; BCSC: Breast Cancer Surveillance Consortium; BOADICEA = Breast and Ovarian Analysis of Disease Incidence and Carrier Estimation Algorithm; DCIS = ductal carcinoma in situ; EPIC: European Investigation into Cancer and Nutrition study; ER = Estrogen receptor; i-Care-BPC3 = Individualized Coherent Absolute Risk Estimation—Breast and Prostate Cancer Cohort Consortium; iCARE-Lit = Individualized Coherent Absolute Risk Estimation—literature based tool; KCPS: Korean Cancer Prevention Study; KREA = tool using Korean incidence, mortality and risk factor distributions with European-ancestry relative risks; KRKR = tool using Korean incidence, mortality and risk factor distributions with Korean relative risks; MWS: Marin Women’s Study; N = number of participants; NHS = Nurses’ Health Study; NR = not reported; PLCO: Prostate, Lung, Colorectal and Ovarian Cancer Screening Trial; ProF-SC: Breast Cancer Prospective Family Study Cohort; PRS = polygenic risk score; SBCSP: Singapore Breast Cancer Screening Project; TC = Tyrer-Cuzick; v = version; WGHS: Women’s Genome Health Study; WHI = Women’s Health Initiative.

**Table 2 cancers-15-01124-t002:** Assessment of risk assessment tools’ validation using metrics for expected/observed rates and trend in observed breast cancer incidence rates.

Study (Country, Age Range)	Model	Proportion of Cohort Well-Validated ^a^	Evidence of Miscalibration (*p*-Value)	Mis- Calibration ^b^	Lower Q Compared to Middle Qs (*p*-Value)	Distinguishes Women in Lowest RG? ^b,c^	Upper Q Compared to Middle Qs (*p*-Value)	Distinguishes Women in Highest RG? ^b,c^	Trend in Observed Rates
Tyrer-Cuzick vs. BCRAT (5-year risk)									
Jantzen 2021, [33] (Canada, 50–69 y)	TC v8.0b	2/4 (18%)	0.045	Yes	<0.001	N/A	<0.001	N/A	Fluctuating
	BCRAT v4	3/4 (84%)	0.035	Yes	<0.001	N/A	<0.001	N/A	Fluctuating
Terry 2019 [31] (USA, Canada, Australia, 20–70 y)	TC v8.0b	2/4 (40%)	<0.001	Yes	<0.001	N/A	<0.001	N/A	Increasing
	BCRAT v4	1/4 (16%)	<0.001	Yes	0.004	N/A	0.004	N/A	Increasing
Tyrer-Cuzick vs. BCRAT (10-year risk)									
Terry 2019 [31], (USA, Canada, Australia, 20–70 y)	BCRAT v4	1/4 (26%)	<0.001	Yes	<0.001	N/A	<0.001	N/A	Increasing
	TC v8.0b	2/4 (42%)	<0.001	Yes	<0.001	N/A	<0.001	N/A	Increasing
Tyrer-Cuzick vs. its variants or other tools (5–6 year risk)									
Choudhury 2020 [35], 5 y risk (UK cohort)	TC v8 (<50 y)	9/10 (90%)	0.074	No	<0.001	Yes	<0.001	Yes	Fluctuating
	iCARE-Lit (<50 y)	10/10 (100%)	0.251	No	0.006	Yes	<0.001	Yes	Fluctuating
	TC v8 (≥50 y)	7/10 (70%)	<0.001	Yes	<0.001	Yes	<0.001	Yes	Fluctuating
	iCARE-Lit (≥50 y)	9/10 (90%)	0.010	Yes	<0.001	Yes	<0.001	Yes	Fluctuating
	iCARE-BPC3 (≥50 y)	9/10 (90%)	0.997	No	<0.001	Yes	<0.001	Yes	Fluctuating
McCarthy 2020 [34], 6 y risk (USA, 40–84 y)	TC v.7	7/10 (70%)	0.002	Yes	<0.001	Yes	<0.001	Yes	Fluctuating
	TC v8.0b	6/10 (60%)	<0.001	Yes	<0.001	Yes	<0.001	Yes	Fluctuating
Tyrer-Cuzick tool variants (10-year risk)									
Brentnall 2018 [38] 10 y risk (USA, 40–75 y)	TC v7.02	2/5 (55%)	<0.001	Yes	<0.001	N/A	<0.001	N/A	Increasing
	TC v7.02 + MD	2/5 (47%)	<0.001	Yes	<0.001	N/A	<0.001	Yes	Increasing
BCRAT vs. its modifications (5-year risk)									
Chay 2012 [32], (Singapore, 50–64 y)	BCRAT v2	0/5 (0%)	<0.001	Yes	0.269	No	0.004	Yes	Fluctuating
	AABCS	3/5 (60%)	<0.001	Yes	0.082	No	<0.001	Yes	Monotonic
Hüsing 2020 [36] (Germany, 20–70 y)	BCRAT v3	10/10 (100%)	0.918	No	<0.001	Yes	0.018	Yes	Fluctuating
	BCRmod	10/10 (100%)	0.227	No	0.002	Yes	<0.001	Yes	Fluctuating
	BCRAT v3 recalibrated	10/10 (100%)	0.324	No	<0.001	Yes	0.011	Yes	Fluctuating
	BCRmod recalibrated	7/10 (70%)	0.007	Yes	<0.001	Yes	<0.001	Yes	Fluctuating
Min 2014 [40] (Republic of Korea, >29–60 y)	BCRAT v2	1/5 (19%)	<0.001	Yes	0.333	No	0.010	Yes	Fluctuating
	AABCS	2/5 (40%)	<0.001	Yes	0.464	No	0.016	Yes	Fluctuating
BCRAT vs. its modifications (10-year risk)									
Chay 2012 [32], (Singapore, 50–64 y)	BCRAT v2	0/5 (0%)	<0.001	Yes	0.253	No	<0.001	Yes	Fluctuating
	AABCS	5/5 (100%)	0.719	No	0.007	Yes	<0.001	Yes	Increasing
BCRAT vs. other risk assessment tools (5-year risk)									
Arrospide 2013 [41] (Spain, 50–69 y)	BCRAT v1	5/5 (100%)	0.289	No	0.599	No	0.004	Yes	Fluctuating
	Chen v1	5/5 (100%)	0.124	No	0.430	No	0.060	No	Fluctuating
Choudhury 2020 [35] (USA cohort, 50–75 y)	BCRAT v3	3/10 (30%)	<0.001	Yes	0.045	Yes	<0.001	Yes	Fluctuating
	iCARE-Lit	6/10 (60%)	<0.001	Yes	<0.001	Yes	<0.001	Yes	Fluctuating
McCarthy 2020 [34] (6-year risk only) (USA, 40–84 y)	BCRAT v4	10/10 (100%)	0.863	No	<0.001	Yes	<0.001	Yes	Fluctuating
	BRCAPRO v2.1–4	9/10 (90%)	0.061	No	<0.001	Yes	<0.001	Yes	Fluctuating
Powell 2014 [30] (USA, >40–80 y)	BCRAT v2 or 3	9/10 (90%)	0.009	Yes	<0.001	Yes	0.003	Yes	Fluctuating
	BRCAPRO v(NR)	4/10 (40%)	<0.001	Yes	0.012	Yes	<0.001	Yes	Fluctuating
Terry 2019 [31] (USA, Canada, Australia, 20–70 y)	BCRAT v4	1/4 (26%)	<0.001	Yes	0.004	N/A	<0.001	N/A	Increasing
	BRCAPRO v2.1–3	0/4 (0%)	<0.001	Yes	<0.001	N/A	<0.001	N/A	Increasing
	BOADICEA v3	2/4 (44%)	<0.001	Yes	<0.001	N/A	<0.001	N/A	Increasing
BCRAT vs. other risk assessment tools (10-year risk)									
Terry 2019 [31], (USA, Canada, Australia, 20–70 y)	BCRAT v4	1/4 (26%)	<0.001	Yes	<0.001	N/A	<0.001	N/A	Increasing
	BRCAPRO v2.1–3	1/4 (7%)	<0.001	Yes	<0.001	N/A	<0.001	N/A	Increasing
	BOADICEA v3	3/4 (66%)	<0.001	Yes	<0.001	N/A	<0.001	N/A	Increasing
Tool comparisons with and without polygenic risk scores (5-year risk)									
Hurson 2021 [29] (UK cohort)	iCARE-Lit (<50 y)	6/10 (60%)	<0.001	Yes	<0.001	Yes	<0.001	Yes	Fluctuating
	iCARE-Lit + PRS (<50 y)	8/10 (80%)	<0.001	Yes	<0.001	Yes	<0.001	Yes	Increasing
	iCARE-Lit (≥50 y)	9/10 (90%)	0.041	Yes	<0.001	Yes	<0.001	Yes	Fluctuating
	iCARE-Lit + PRS (≥50 y)	9/10 (90%)	0.004	Yes	<0.001	Yes	<0.001	Yes	Fluctuating
	iCARE-BPC3 (≥50 y)	10/10 (100%)	0.020	Yes	<0.001	Yes	<0.001	Yes	Fluctuating
	iCARE-BPC3 + PRS (≥50 y)	10/10 (100%)	0.002	Yes	<0.001	Yes	<0.001	Yes	Fluctuating
Other risk assessment tools (5-year risk)									
Jee 2020 [37] (Republic of Korea)	KREA (<50 y)	5/10 (50%)	0.022	Yes	<0.001	Yes	<0.001	Yes	Fluctuating
	KRKR (<50 y)	4/10 (40%)	0.383	No	<0.001	Yes	<0.001	Yes	Fluctuating
	KREA (≥50 y)	6/10 (60%)	0.341	No	0.002	Yes	0.160	No	Fluctuating
	KRKR (≥50 y)	3/10 (30%)	0.127	No	0.005	Yes	0.222	No	Fluctuating
Li 2018 [39] (USA, 50–79 y)	ER-	9/10 (90%)	0.044	Yes	0.810	No	0.380	No	Fluctuating
	ER+	9/10 (90%)	<0.001	Yes	<0.001	Yes	<0.001	Yes	Fluctuating
Min 2014 [40] (Republi of Korea, >29–60 y)	Original Korean tool	1/5 (20%)	<0.001	Yes	0.439	No	0.356	No	Fluctuating
	Updated Korean tool	2/5 (40%)	<0.001	Yes	0.640	No	0.022	Yes	Fluctuating

^a^ Evaluation of well-validated risk groups is based on the corresponding 95% confidence intervals of point estimates including 1; ^b^ Based on *p*-value of <0.05 for statistical significance, ^c^ To ensure comparability of findings, if >25% of the study cohort was in the highest or/and the lowest risk groups, *p*-values were reported but were not used to determine if the tool distinguished women in highest or lowest risk groups. Abbreviations: AABCS = Asian American Breast Cancer Study; BCRAT = Breast cancer risk assessment tool; BOADICEA = Breast and Ovarian Analysis of Disease Incidence and Carrier Estimation Algorithm; ER = Estrogen receptor; i-Care-BPC3 = Individualized Coherent Absolute Risk Estimation—Breast and Prostate Cancer Cohort Consortium; iCARE-Lit = Individualized Coherent Absolute Risk Estimation—literature based tool; KREA = tool using Korean incidence, mortality and risk factor distributions with European-ancestry relative risks; KRKR = tool using Korean incidence, mortality and risk factor distributions with Korean relative risks; NHS = Nurses’ Health Study; PRS = polygenic risk score; Q: quartile; TC = Tyrer-Cuzick; v = version; v: version; WHI = Women’s Health Initiative; y: years.

**Table 3 cancers-15-01124-t003:** Summary of risk of bias of included breast cancer risk assessment tool studies for breast cancer calibration outcomes. low risk is green, high risk is red and undetermined is orange.

Study	RAT	Cohort	Year	Outcome	Participants	Predictors	Outcome	Analysis ^a^	Overall RoB
Hurson 2021 [29]	iCARE BPC3	UK Biobank	5	Invasive or DCIS	LR	LR	U	HR	HR
Hurson 2021 [29]	iCARE BPC3 LR PRS	UK Biobank	5	Invasive or DCIS	LR	U	U	HR	HR
Hurson 2021 [29]	iCARE Lit	UK Biobank	5	Invasive or DCIS	LR	LR	U	HR	HR
Hurson 2021 [29]	iCARE Lit LR PRS	UK Biobank	5	Invasive or DCIS	LR	U	U	HR	HR
Hurson 2021 [29]	iCARE Lit	WGHS	5	Invasive or DCIS	LR	U	U	HR	HR
Hurson 2021 [29]	iCARE Lit LR PRS	WGHS	5	Invasive or DCIS	LR	U	U	HR	HR
Jantzen 2021 [33]	TC v8	CARTaGENE	5	Invasive	LR	LR	U	HR	HR
Jantzen 2021 [33]	BCRAT v4	CARTaGENE	5	Invasive	LR	LR	U	HR	HR
McCarthy 2020 [34]	TC v7	NWH	6	Invasive	HR	LR	U	HR	HR
McCarthy 2020 [34]	TC v8.0b	NWH	6	Invasive	HR	LR	U	HR	HR
McCarthy 2020 [34]	BCRAT v4	NWH	6	Invasive	LR	LR	U	HR	HR
McCarthy 2020 [34]	BRCAPRO v2.1HR4	NWH	6	Invasive	HR	LR	U	HR	HR
Choudhury 2020 [35]	TC v8	GS	5	Invasive	LR	U	U	HR	HR
Choudhury 2020 [35]	iCARE Lit	GS	5	Invasive	LR	U	U	HR	HR
Choudhury 2020 [35]	iCARE BPC3	GS	5	Invasive	LR	U	U	HR	HR
Choudhury 2020 [35]	BCRAT v3	PLCO	5	Invasive	LR	LR	U	HR	HR
Choudhury 2020 [35]	iCARE Lit	PLCO	5	Invasive	LR	LR	U	HR	HR
Hüsing 2020 [36]	BCRAT v3	EPICHRGermany	5	Invasive	HR	U	HR	HR	HR
Hüsing 2020 [36]	BCRmod	EPICHRGermany	5	Invasive	LR	U	HR	HR	HR
Hüsing 2020 [36]	BCRAT v3 recalibrated	EPICHRGermany	5	Invasive	HR	U	HR	HR	HR
Hüsing 2020 [36]	BCRmod recalibrated	EPICHRGermany	5	Invasive	LR	U	HR	HR	HR
Jee 2020 [37]	KREA	KCPSHRII Biobank	5	Invasive	LR	LR	U	HR	HR
Jee 2020 [37]	KRKR	KCPSHRII Biobank	5	Invasive	LR	LR	U	HR	HR
Terry 2019 [31]	BCRAT v4	ProFHRSC	5	Invasive	HR	HR	HR	HR	HR
Terry 2019 [31]	BRCAPRO v2.1HR3	ProFHRSC	5	Invasive	LR	HR	HR	HR	HR
Terry 2019 [31]	TC v8.0b	ProFHRSC	5	Invasive	LR	HR	HR	HR	HR
Terry 2019 [31]	BOADICEA v3	ProFHRSC	5	Invasive	LR	HR	HR	HR	HR
Terry 2019 [31]	BCRAT v4	ProFHRSC	10	Invasive	HR	HR	HR	HR	HR
Terry 2019 [31]	BRCAPRO v2.1HR3	ProFHRSC	10	Invasive	LR	HR	HR	HR	HR
Terry 2019 [31]	TC v8.0b	ProFHRSC	10	Invasive	LR	HR	HR	HR	HR
Terry 2019 [31]	BOADICEA v3	ProFHRSC	10	Invasive	LR	HR	HR	HR	HR
Brentnall 2018 [38]	TC v7.02	KPWHRBCSC	10	Invasive	LR	HR	U	HR	HR
Brentnall 2018 [38]	TC v7.02 LR BD	KPWHRBCSC	10	Invasive	LR	HR	U	HR	HR
Li 2018 [39]	ERHR	WHI	5	Invasive	LR	U	HR	HR	HR
Li 2018 [39]	ERLR	WHI	5	Invasive	LR	U	HR	HR	HR
Min 2014 [40]	BCRAT v2	WHC CGH	5	Invasive	HR	LR	U	HR	HR
Min 2014 [40]	AABCS	WHC CGH	5	Invasive	HR	LR	U	HR	HR
Min 2014 [40]	Original Korean tool	WHC CGH	5	Invasive	HR	LR	U	HR	HR
Min 2014 [40]	Updated Korean tool	WHC CGH	5	Invasive	HR	LR	U	HR	HR
Powell 2014 [30]	BCRAT v2 or 3	MWS	5	Invasive	HR	HR	U	HR	HR
Powell 2014 [30]	BRCAPRO v(NR)	MWS	5	Invasive	LR	HR	U	HR	HR
Arrospide 2013 [41]	BCRAT v1	SCHRBCSP	5	Invasive	LR	LR	HR	HR	HR
Arrospide 2013 [41]	Chen v1	SCHRBCSP	5	Invasive	LR	HR	HR	HR	HR
Chay 2012 [32]	BCRAT v2	SBCSP	5	Invasive	LR	HR	U	HR	HR
Chay 2012 [32]	AABCS	SBCSP	5	Invasive	LR	HR	U	HR	HR
Chay 2012 [32]	BCRAT v2	SBCSP	10	Invasive	LR	HR	U	HR	HR
Chay 2012 [32]	AABCS	SBCSP	10	Invasive	LR	HR	U	HR	HR

^a^ Note: Items 4.5, 4.8 and 4.9 omitted as they are signalling questions for model development and not validation; Key to domain and overall rating: High risk of bias: indicated as ‘HR‘; low risk of bias: indicated as ‘LR’; unclear risk of bias: indicated as ‘U’ Abbreviations: AABCS = Asian American Breast Cancer Study; BCRAT = Breast cancer risk assessment tool; BOADICEA = Breast and Ovarian Analysis of Disease Incidence and Carrier Estimation Algorithm; DCIS = ductal carcinoma in situ; ER = Estrogen receptor; i-Care-BPC3 = Individualized Coherent Absolute Risk Estimation—Breast and Prostate Cancer Cohort Consortium; iCARE-Lit = Individualized Coherent Absolute Risk Estimation—literature based tool; KREA = tool using Korean incidence, mortality and risk factor distributions with European-ancestry relative risks; KRKR = tool using Korean incidence, mortality and risk factor distributions with Korean relative risks; N = number of participants; NHS = Nurses’ Health Study; NR = not reported; ROB: risk of bias; PRS = polygenic risk score; TC = Tyrer-Cuzick; v = version; WHI = Women’s Health Initiative.

## Data Availability

The data supporting the findings of this work are available in the cited articles and in the manuscript’s Appendix A.

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
