# Peer review of "Breast Cancer Risk Assessment Tools for Stratifying Women into Risk Groups: A Systematic Review"

_cancers, 2023, doi:10.3390/cancers15041124_

Round 1

Reviewer 1 Report

In this systematic review, the authors aimed to assess how accurately breast cancer risk assessment tools (Gail, Tyrer-Cuzick, etc) can group women eligible for screening within a population into risk groups.  A robust systematic analysis was performed by the authors. They concluded that most tools identified a risk group with higher rates of observed cancers but few tools identified lower-risk groups across different settings. The authors reported that all tools demonstrated a high risk of bias. This is a relevant study and likely of interest to the readership of Cancers. There are some limitations to the manuscript listed below:

Major comments:

-       Very well written but could try to summarize a lot of the data and reorganize to make the paper easier to read.

-       Would consider writing the abstract as a narrative paragraph 

-       Please clarify “Comparison: different tool” – what do you mean by that in the abstract

-       Please explain why did you exclude articles limited to women undergoing diagnostic breast imaging, specific ethnic groups or women with a high risk of breast cancer

-       Add PRISMA diagram to methods 

-       In your conclusions or discussion, please include what do you recommend to do based on this information and what are your next steps 

Minor comments: 

-       Rephrase or define “goodness of fit” in the abstract 

-       Would add a marginal row to table 1 with the number of articles, participants, etc 

-       Would explain figure 1 more clearly (in the paper or within the figire) and try to simplify it, it is hard to follow. 

Author Response

Monday, 30th January 2023

Dear reviewer,

Thank you for taking the time to review our manuscript entitled ‘Breast cancer risk assessment tools for stratifying women in risk groups: a systematic review’ submitted to the journal ‘Cancers’ and providing comments; we hope we have addressed these. Please find our responses below.

Kind regards,

Dr Louiza S Velentzis (on behalf of all authors)

Major comments

  1. Very well written but could try to summarize a lot of the data and reorganize to make the paper easier to read.

Response:
We thank the reviewer for their comment. The article is structured by having different sub-sections in the results and discussion accompanied by relevant titles to provide more clarity for the reader. The reviewer has not provided enough detail to enable us to make any further changes.

  1. Would consider writing the abstract as a narrative paragraph

Response:
We have followed the journal’s specifications regarding how the abstract should be structured, and therefore have not converted the abstract into a narrative.

  1. Please clarify “Comparison: different tool” – what do you mean by that in the abstract

Response:
Included articles had to have one risk assessment tool compared to a second, different risk assessment tools, with both applied to the same cohort. To clarify this point we have amended the abstract as indicated by the bold text below:

“Methods: Population: asymptomatic women aged ≥40 years; Intervention: questionnaire-based risk assessment tool (incorporating breast density and polygenic risk where available); Comparison: different tool applied to the same population; Primary outcome: breast cancer incidence; Scope: external validation studies identified from databases including Medline and Embase (period 1/1/2008-20/07/2021).”

  1. Please explain why did you exclude articles limited to women undergoing diagnostic breast imaging, specific ethnic groups or women with a high risk of breast cancer.

Response:
This article focuses on the application of risk assessment tools in the general female population undergoing mammographic screening. The groups excluded are sub-groups of the population. Under the eligibility criteria in the methods section we have amended the following sentences in the manuscript as indicated:

“The current analysis was confined to articles comparing breast cancer risk assessment tools on the same study cohort; cohorts had to consist of asymptomatic women undergoing population mammographic screening. We excluded articles limited to cohorts of women undergoing diagnostic breast imaging, specific ethnic groups or women with high risk of breast cancer as these represent sub-groups of the screened population.”

  1. Add PRISMA diagram to methods.

Response:
The PRISMA diagram has now been added to the methods section.

  1. In your conclusions or discussion, please include what do you recommend to do based on this information and what are your next steps.

Response:

Based on the findings of the systematic review we have made a number of recommendations in various sections in our discussion, as well as potential future work, as indicated below. Therefore, we have not made changes in response to this comment:

  • In section 4.3 (applicability and model performance) we state:

“For full transparency it is recommended that authors provide the specific version of the risk assessment tools used including the software package, all predictors offered by that version and used in the study being reported.”

  • In section 4.4 (risk of bias) we state:

“We recommend that future studies consider including information on how missing data are managed, as this would improve comparability between studies and help recognise the challenges of applying risk assessment tools to different settings and study populations.”

Also:

“We also recommend more standardised and transparent reporting of risk assess-ment tools, using the ‘Transparent Reporting of a multivariable prediction model for Individual Prognosis or Diagnosis’ (TRIPOD) statement published in 2015 [49].”

  • In section 4.2 (comparison with other published work) we stated the wording shown below. We have also mended this sentence to explicitly indicate that this is a recommendation:

“We recommend the use of observed rates of incident breast cancer according to tool-determined risk groups as it provides a better quantitative assessment of discrimination for this purpose, informing consideration of interventions that might target women at different thresholds of risk across the risk spectrum.”

In terms of next steps, in the conclusion we have proposed further work evaluating the use of tools based on a smaller number of risk factors; these may be more easily implemented in a setting such as breast cancer screening.

                “..in parallel, work is required to understand the relative utility of more parsimonious tools that may achieve similar outcomes while markedly reducing the impost of risk assessment on women and health services.”

Minor comments:

  1. Rephrase or define “goodness of fit” in the abstract

Response:

The reviewer is likely referring to the following sentence in the abstract.

“We assessed goodness-of-fit (calibration) between expected and observed cancers and compared observed cancer rates by risk group. Risk of bias was assessed with PROBAST.”

We have used the term ‘goodness-of-fit’ but also the term ‘calibration’ in brackets. Since calibration may be the most conventional term we have amended the sentence as follows:

“We assessed calibration (goodness-of-fit) between expected and observed cancers and compared observed cancer rates by risk group. Risk of bias was assessed with PROBAST.”

  1. Would add a marginal row to table 1 with the number of articles, participants, etc

Response:

We already note the number of articles included in the systematic review in the 2nd paragraph of the ‘Results’ section. As this is review does not include a meta-analysis, providing the total number of participants in all studies combined would not be informative, and would potentially be misleading. Hence, we have not added an additional line in Table 1.

  1. Would explain figure 1 more clearly (in the paper or within the figire) and try to simplify it, it is hard to follow.

Response:

We have now amended the caption of this figure (now appearing as figure 2) to provide more information for the reader as indicated by the bold text below.

Figure 2. Absolute risk calibration and observed rate of incident breast cancer by 5-year risk. The three groups shown are: A) Tyrer-Cuzick vs. BCRAT or other tool comparisons; B) BCRAT vs. BCRAT modifications; C) BCRAT vs. other risk assessment tools. Plots are then presented according to first author name. (The number of data points in each graph is determined by the number of risk groups that were reported in each study. To assist with comparison of studies, the x-axis shows the percentile distribution of groups being reported, with data points shown for the mid-points of each group. Red squares show the ‘expected over observed’ ratio for each risk group (with 95% confidence intervals shown), indicating calibration between expected and observed cancers at a risk group level. Blue circles show the corresponding observed rate of breast cancers within the study group, indicating the gradient of rates across the risk groups (expected to increase from left to right in accordance with increases in estimated breast cancer risk). Italic font indicates the risk tool being assessed, with the study cohort abbreviation also shown).

Reviewer 2 Report

It was a nice and detailed study, especially in terms of statistical analysis. Congratulations.

Author Response

Monday, 30th January 2023

Dear reviewer,

Thank you for taking the time to review our manuscript entitled ‘Breast cancer risk assessment tools for stratifying women in risk groups: a systematic review’ submitted to the journal ‘Cancers’ and providing comments; we hope we have addressed these. Please find our responses below.

Kind regards,

Dr Louiza S Velentzis (on behalf of all authors)

Reviewer 2

It was a nice and detailed study, especially in terms of statistical analysis. Congratulations.

Response:
We thank the reviewer for their comment; much appreciated.

Reviewer 3 Report

This work is very important for many science people how to evaluate the data properly.It looks like a tutorial article. Thanks for your great work.

Author Response

Monday, 30th January 2023

Dear reviewer,

Thank you for taking the time to review our manuscript entitled ‘Breast cancer risk assessment tools for stratifying women in risk groups: a systematic review’ submitted to the journal ‘Cancers’ and providing comments; we hope we have addressed these. Please find our responses below.

Kind regards,

Dr Louiza S Velentzis (on behalf of all authors)

Reviewer 3

This work is very important for many science people how to evaluate the data properly. It looks like a tutorial article. Thanks for your great work.

Response:
We thank the reviewer for their comment. Due to the inclusion of various metrics to evaluate the data, we have provided a certain amount of detail to enable clarity for a wide audience.

Reviewer 4 Report

The authors present a systematic review of risk prediction tools that stratify women in different cohorts regarding breast cancer risk. The work is interesting regarding its content and the aim. It is important to advance the available evidence regarding the existence of predictive tools in the context of predictive medicine. The authors did a good job in organizing all this information and should be congratulated about that. But this also creates an issue. I believe a main issue of the MS is that it is too dense in its content with a lot of information making it difficult to read or difficult to follow the results and discussion so the reader to extract the main outcomes of this work. For example, a clear comparison between the tools is missing. Although there are comments in discussion section etc. it is not organized into a single paragraph what each tool provides and what it lacks (perhaps a summarization of table 2). Also, a question that is partially presented within the text. How "homogenous" in their content and the way the perform the evaluation are the analyzed tools? Also, how each group is defined as high-medium-low risk for each tool?  Moreover, any differences among these tools (for example in their performance) regarding the type of breast cancer that they best predict(invasive, non-invasive, Malignant phyllodes tumors, HR+/-). Overal, what is the take home message from this work, what is the main contribution. It is something that should be better presented

Some additional comments: 

1)Simple summary could be improved including some additional information such as that the importance of early diagnosis for breast cancer. 

2) "The benefits and harms of breast diagnosis maybe better balanced through a risk-stratified approach." Not sure what the authors mean by that. Probably the ref 5 but is any tool considerign the "harm" as it is described later? 

3) Why articles limited to specific ethnic groups where excluded? (eligibility criteria). Considering also that 10/13 studies were in Europe and North America wouldn't this create any issue of the predictive value for other ethnicities? 

4) Methods section is too long and could be shortened. For example there is no need to explain in details PROBAST. 

5) I would suggest the PRISMA flow to be included in the main body of the text and not in supplementary. 

6) Table 1. Needs to be simplified or edited to be in a different page with horizontal layout

7) Maybe some sections in discussion could be moved to introduction "Our review of studies comparing multiple breast cancer risk assessment tools...accurete estimate". 

8) Maybe results and discussion would be better presented as one section to help the reading flow of the MS. I leave it to the discretion of authors and the editor.

9) "What does this logic mean for how we evaluate risk assessment tools for the purposes

of potential risk-based population breast screening?" These kind of phrases should be avoided within a scientific text and a text with a more "sterile" scientific writing should be followed. 

The main outcome from the study is that all these tools, generally, are capable to stratify women in high risk cohorts but not is lower risk. What could be improved on that matter?  

I mark it with major changes although most of the comments are minor ones. I believe though that the authors will need time to edit their work so I mark it with major revisions. 

Author Response

Monday, 30th January 2023

Dear reviewer,

Thank you for taking the time to review our manuscript entitled ‘Breast cancer risk assessment tools for stratifying women in risk groups: a systematic review’ submitted to the journal ‘Cancers’ and providing comments; we hope we have addressed these. To enable a more detailed response, please note that the annotations a-e below are added by the authors, as the first set of comments were provided in a single paragraph. Please find our responses below.

Kind regards,

Dr Louiza S Velentzis (on behalf of all authors)

Reviewer 4

  1. The authors present a systematic review of risk prediction tools that stratify women in different cohorts regarding breast cancer risk. The work is interesting regarding its content and the aim. It is important to advance the available evidence regarding the existence of predictive tools in the context of predictive medicine. The authors did a good job in organizing all this information and should be congratulated about that. But this also creates an issue. I believe a main issue of the MS is that it is too dense in its content with a lot of information making it difficult to read or difficult to follow the results and discussion so the reader to extract the main outcomes of this work. For example, a clear comparison between the tools is missing. Although there are comments in discussion section etc. it is not organized into a single paragraph what each tool provides and what it lacks (perhaps a summarization of table 2).

Response:

This is a complex paper, and we understand the reviewer’s point of view that this would be a helpful summary to assist the reader. However, summarizing the information for each tool separately will, in our opinion, be more confusing as results often vary when applied to different cohorts/settings, with different studies using some or all of the risk factor data, and depending on how results were presented in the source publications (e.g. four vs five vs ten risk groups). In a body of literature which revealed itself to use a wide variety of methods, our approach was to use one metric at a time to systematically and clearly demonstrate how each tool performed in a single cohort and setting, and what can be learned.

  1. Also, a question that is partially presented within the text. How "homogenous" in their content and the way the perform the evaluation are the analyzed tools?

Response:

We are unsure of the exact points the reviewer is raising here. If the reviewer is referring to the predictors within each tool, we have stated in the results that tools varied in terms of the number of predictors included which does lead to some heterogeneity between tools.. Furthermore, as described in the discussion section, there are differences between the studies in which the tools are applied. For this reason, we did not apply a single reductive approach to our comparisons, as there is the potential for biased interpretation of studies through selecting which metrics to value (e.g. AUC values, or calibration on its own, without regard for observed rates). Instead, we systematically assessed a range of metrics and argue that different assessments should be considered in consort. We have made this clearer by adding the following text (indicated in bold) to the first sentence of Discussion paragraph 2: ‘We did not apply a single metric to compare tools because the interpretation and value of each metric depends on how the risk assessment tool might be used.’

We cannot interpret the second part of this question, as the grammar is not coherent (‘the way the perform the evaluation’)

  1. Also, how each group is defined as high-medium-low risk for each tool?

Response:

Although these risk groups are mentioned within the Methods, section B3 and 4, where we described how high risk and low risk groups can be distinguished from moderate risk groups, we agree with the reviewer that it would be useful to define these groups earlier on. We have therefore added the following text in the methods, at the end of section 2.5, as shown in bold:

“We also identified high, moderate and low risk groups for each tool in each cohort. These groups were dependent on the number of quantiles the cohort of interest was divided into and whether they had the equivalent number of participants in each one. In general, when the cohort was divided in equal quartiles or deciles, we assumed the high-risk group corresponded to quartile 5 or deciles 9 and 10, the low-risk group corresponded to quartile 1 or deciles 1 and 2 while moderate risk groups correspond to the remaining quantiles (quartiles 2-4 or deciles 3-8).”

  1. d) Moreover, any differences among these tools (for example in their performance) regarding the type of breast cancer that they best predict (invasive, non-invasive, Malignant phyllodes tumors, HR+/-).

Response:

Table 1 includes the type of breast cancer(s) predicted by the risk assessment tools applied in each included study, indicating that all studies predicted invasive breast cancers with the exception of Hurson et al, 2021 which also predicted DCIS.

  1. Overal, what is the take home message from this work, what is the main contribution. It is something that should be better presented.

Response:

We believe we have included the main take home messages in the first two sentences of our conclusion. Namely that i) there are a number of risk assessment tools that are effective in assigning women to breast cancer risk groups as determined by various metrics of tool performance and that ii) the best metrics to assess a tool depends on how a risk tool is going to be used.

Some additional comments:

1)Simple summary could be improved including some additional information such as that the importance of early diagnosis for breast cancer.

Response:

We agree with the reviewer, noting that this would then exceed the journal’s specified word count for this section of 150 words. We have provided additional text in the summary as indicated below in bold (subject to journal approval to extend the word count):

Early detection of breast cancer in asymptomatic women through screening is an important strategy in reducing the burden of breast cancer. In current organized breast screening programs, age is the predominant risk factor. Breast cancer risk assessment tools are numerical models that can combine information on various risk factors to estimate the risk of being diagnosed with breast cancer within a certain time period. These tools could be used to offer risk-based screening.”

2) "The benefits and harms of breast diagnosis maybe better balanced through a risk-stratified approach." Not sure what the authors mean by that. Probably the ref 5 but is any tool considerign the "harm" as it is described later?

Response:

The reviewer has misread the first line of the abstract which refers to the benefits and harms of breast cancer screening, not of breast diagnosis. Risk assessment tools could help identify groups of women at different levels of breast cancer risk, with each group offered a breast screening protocol that would optimise the benefits of screening (i.e. identifying cancers at an early stage) while minimising harms (such as overdiagnosis, false positives and interval cancers).

3) Why articles limited to specific ethnic groups where excluded? (eligibility criteria). Considering also that 10/13 studies were in Europe and North America wouldn't this create any issue of the predictive value for other ethnicities?

Response:

We excluded studies focusing on ethnic groups, because they are subgroups of the general population of the country in which these studies took place and therefore not representative of this population. We have now amended the relevant sentences in the methods section to clarify this point further in the eligibility criteria as shown in bold below:

“The current analysis was confined to articles comparing breast cancer risk assessment tools on the same study cohort; cohorts had to consist of asymptomatic women undergoing population mammographic screening. We excluded articles limited to cohorts of women undergoing diagnostic breast imaging, specific ethnic groups or women with high risk of breast cancer as these represent sub-groups of the screened population.”

4) Methods section is too long and could be shortened. For example there is no need to explain in details PROBAST.

Response:

We have provided sufficient detail in the methods section to provide context for the reader to better understand the results, acknowledging that the audience have different scientific backgrounds. In this light, we have provided information about the domains of PROBAST as we refer to these in our results. A reader who is familiar with this tool is welcomed to skip this section.

5) I would suggest the PRISMA flow to be included in the main body of the text and not in supplementary.

Response:

We agree with the reviewed and have moved the PRISMA flow diagram in the main article.

6) Table 1. Needs to be simplified or edited to be in a different page with horizontal layout

Response:

Table 1 was submitted to the journal in a horizontal i.e. landscape layout, however, the journal converted the draft manuscript sent to reviewers to a vertical layout. We will kindly request from the journal for the table be reverted to its original landscape format so that the information appears less squashed and confirm this.

7) Maybe some sections in discussion could be moved to introduction "Our review of studies comparing multiple breast cancer risk assessment tools...accurete estimate".

Response:

The paragraph mentioned by the reviewer summarises the findings of our systematic review and therefore cannot be moved to the introduction.

8) Maybe results and discussion would be better presented as one section to help the reading flow of the MS. I leave it to the discretion of authors and the editor.

Response:

We believe having separate results and discussion sections provide more clarity for the reader and follow best practice in terms of separating findings from inferences.

9) "What does this logic mean for how we evaluate risk assessment tools for the purposes of potential risk-based population breast screening?" These kind of phrases should be avoided within a scientific text and a text with a more "sterile" scientific writing should be followed.

Response:

We have amended the specific sentence as indicated below:

“The issues mentioned above have potential consequences for how risk assessment tools should be evaluated in relation to risk-based population breast screening.”

10) The main outcome from the study is that all these tools, generally, are capable to stratify women in high risk cohorts but not is lower risk. What could be improved on that matter? 

Response:

The question is beyond the scope of this systematic review.

Round 2

Reviewer 4 Report

As it was noted from the first evaluation there were some issues mostly minor that could be adressed. The authors presented an updated version of their work and more or less adressed the comments raised initialy. The manuscript can be accepted.